# Effect of Black Tea Infusion on Physicochemical Properties, Antioxidant Capacity and Microstructure of Acidified Dairy Gel during Cold Storage

**DOI:** 10.3390/foods9060831

**Published:** 2020-06-25

**Authors:** Han Chen, Haotian Zheng, Margaret Anne Brennan, Wenpin Chen, Xinbo Guo, Charles Stephen Brennan

**Affiliations:** 1Department of Wine, Food and Molecular Biosciences, Lincoln University, Lincoln 7647, New Zealand; jackchen.chen@lincolnuni.ac.nz (H.C.); margaret.brennan@lincoln.ac.nz (M.A.B.); 2Department of Food, Bioprocessing and Nutrition Sciences, Southeast Dairy Foods Research Center, North Carolina State University, Raleigh, NC 27695, USA; 3Dairy Innovation Institute, California Polytechnic State University, San Luis Obispo, CA 93407, USA; 4Tea Science Department, College of Horticulture, South China Agricultural University, Guangzhou 510642, China; michaelcwp@163.com; 5School of Food Science of Engineering, South China University of Technology, Guangzhou 510641, China; guoxinbo@scut.edu.cn

**Keywords:** black tea, acidified dairy gel, textural property, antioxidant capacity, microstructure

## Abstract

The impacts of black tea infusion on physicochemical properties, antioxidant capacity and microstructure of stirred acidified dairy gel (ADG) system have not been fully explored. These impacts were studied during a 28-day cold storage (4 °C) period to explore the feasibility and technical boundaries of making acidified dairy gels in which black tea infusion (BTI) is incorporated. Reconstituted skim milks containing different proportions of BTI were acidified by GDL (glucono-δ-lactone) at 35 °C for making ADG systems. Both textural properties and structural features were characterized; antioxidant capacity was determined through three assays. They are (1) free radical scavenging ability by DPPH (2,2-diphenyl-1-picrylhydrazyl) assay; (2) ABTS [2,2′-azino-bis-(3-ethylbenzothiazoline-6-sulphonic acid)] assay and (3) ferric reducing antioxidant power (FRAP) assay. The microstructure of the ADGs was observed using SEM (scanning electron microscopy) and CLSM (confocal laser scanning microscopy). Results showed that BTI significantly increased the antioxidant capacity of the gel systems and the gel containing 15% BTI was as stable as the control gel in terms of syneresis rate. However lower phase stability (higher syneresis rate) was observed in the ADG with a higher portion of BTI (30% to 60%). The microstructure of the ADGs observed may explain to the phase stability and textural attributes. The results suggested that tea polyphenols (TPs) improved antioxidant capacity in all samples and the interactions between BTI and dairy components significantly altered the texture of ADGs. Such alterations were more pronounced in the samples with higher proportion of BTI (60%) and/or longer storage time (28 days).

## 1. Introduction

Free radicals, by-products of metabolism, such as reactive oxygen species (ROS) are constantly being generated in our body such as hydroxyl radical, superoxide radical, hydrogen peroxide and lipid peroxides [1]. Normally, there is an antioxidant defence system in our body, comprising of several enzymes such as iron-dependent catalase, superoxide dismutase (copper/zinc and manganese-dependent) and selenium-dependent glutathione peroxidase to detoxify these free radicals [2]. However, when there is large number of free radicals, there is a disorder between the generation and removal of free radicals in the body, in which case oxidative stress will occur. This may result in oxidative damage to cellular metabolism and biomolecules, and create the onset of many chronic diseases related to aging such as cardiovascular disease, diabetic disease, neurodegenerative diseases or even cancer [3]. As antioxidants play important roles in preventing or inhibiting oxidation of cellular components, adequate intake of these compounds is beneficial to protect cells from oxidative damages. In this regard, extracts of many polyphenol-rich plants or herbs, such as tea, are used more often either as additive in food industry or consumed directly as a natural source of antioxidants [4].

Tea extracts are rich in phenolic compounds [5], these components including flavonoids are considered as antioxidants [6,7]. Yoghurt has been consumed as a healthy food for a long time since 6000 BCE in central Asia [8] due to its nutritional properties, taste and health benefits as results of fermentation of lactic acid bacteria. Acidified dairy protein gels (ADG) induced by glucono-δ-lactone (GDL) as robust model systems have been commonly used in research for studying ingredient functionalities in yogurt like gels and for studying structure and texture features of such gels [9,10,11,12]. To explore the dairy components-tea infusion interactions in an acidic environment, GDL induced gels were used in this study to remove the influence of unnecessary impacts of live culture [13,14].

Although different analytical methods for antioxidant capacity may lead to different results for the same antioxidant [6], previous study showed that supplementing 5–15% of green tea in yogurt may result up to 31-fold higher radical scavenging activity compared to the control yogurt [15]. Muniandy and co-authors reported that during refrigeration storage condition, antioxidant activity for tea enriched yogurt is stable and it remained constant as shown in a study [2]. However, this study only showed comparison of antioxidant activity between blank control yogurt and tea infusion enriched yogurts. It is not clear that if there is any change of antioxidant activity of the original tea infusion when it incorporated in the dairy protein gel system. In another study, the authors showed that there is nearly no change of ferric reducing antioxidant power of the green tea extract either in yogurt or in its original state (based on back calculated values) [15]. However, regarding to black tea extract, such information is unknown and the situation may not necessarily as same as of green tea extract because black tea infusion contains more than 20 times of gallic acid than that of green tea [5]. It was found that radical scavenging activity of the black tea was lowered by adding milk [16]. Such phenomenon might be attributed to the interaction between gallic acid and milk proteins [17], because research showed that interactions between flavonoids and proteins affect their antioxidant capacity [18]. Ryan and Petit also concluded that the addition of neutral pH milk components may reduce total antioxidant activity of black tea [19]. In general, casein micelles and whey proteins can bind to tea catechins [20,21] via hydrogen bonds between peptide carbonyl and phenolic hydroxyl [22]. The unique structure of protein–catechin complex may reduce the bioaccessibility of the original tea catechin [23]. The phenolic compound composition is different between green and black tea [5]. The later has relatively lower total phenolic content [5]. Besides gallic acid-protein interaction, the lower content of phenolic compounds may be another major reason explaining why black tea added yogurt showed the lower DPPH scavenging activity than green tea yogurt [2]. However, it is still not clear that if the original antioxidant activity of black tea fusion may be affected after incorporating in acid dairy protein gel.

In a relatively recent review article, the authors consolidated previously published results regarding the functionality of polyphenols interacting with milk proteins and pointed out that some of these findings are conflicted [24]. The authors therefore suggested more research is needed for understanding the true mechanism of interaction between phenolic compounds and dairy matrices. Numerous studies have been done on the mixture systems such as milk–tea infusion systems [16,19,21,22,25,26] and dairy gel–tea infusion systems [2,15,27,28,29,30,31]. Regarding to the topic of tea infusion enriched yogurt, many studies focused on the impact of tea fortification on the antioxidant activity and microbial growth [15,20,21,22,29,31]. Few efforts have been done to investigate the impact of storage time on key characteristics of tea infusion enriched ADG systems. Rheology and texture analyser techniques may be useful tools for characterizing textural changes of semi-solid foods such as yogurt gels [32,33]. Najgebauer-Lejko and co-workers found that the mechanical properties and syneresis rate of tea infusion enriched yogurts depended on the type of tea. The study concluded that green tea incorporation resulted relatively favourable texture and lower syneresis compared to Pu-erh tea infusion [27]. However, black tea infusion was not included in that study. It is clear that the overall antioxidant activity of tea–yogurt system is positively correlated with the volume fraction of tea infusion incorporated in the yogurt gel [15]. However, from application point of view, it is still unclear about what is the highest supplementation rate of tea infusion that a stirred ADG can be tolerant without compromising phase stability and textural characteristics. To the best of our knowledge and according to the aforementioned referenced research, the tea-enriched stirred ADG has never been studied with higher supplementation rates of tea (>15%, *w*/*w*).

Based on the mentioned research questions, the current study aimed to investigate the impact of black tea supplementation on phase stability, textual characteristics, and antioxidant capacity of stirred ADG made from reconstituted milk system. The phase stability of ADG was investigated through cold storage (4 °C for up to 28 days). Tea infusion were incorporated in the ADG systems at different volume fractions (15–60%, *w*/*w*) for testing the formulation boundary. This work provides an in-depth understanding of interaction between ADG network and black tea infusion. Such knowledge may contribute to the development of dairy products with enhanced nutritional benefits and without compromised texture.

## 2. Materials and Methods

### 2.1. Materials

Skimmed milk powder (SMP, fat 1.2%, protein 33.0%, carbohydrate 54%, sodium 0.39%, calcium 1.24%, Pams, Auckland, New Zealand) and bagged black tea (premium Ceylon tea, Dilmah Co., Peliyagoda, Sri Lanka) were purchased from local supermarket (Christchurch, New Zealand). DPPH (2,2-Diphenyl-1-picrylhydrazyl), ABTS (2,2-Diphenyl-1-picrylhydrazyl), TPTZ (2,4,6-Tris(2-pyridyl)-s-triazine), gallic acid were purchased form Sigma-Aldrich Co., New York, NY, USA). Trolox (6-hydroxy-2,5,7,8-tetramethylchloromane-2-carboxylic acid) was purchased from Acros Organic (NS, New York, NY, USA). Phosphate buffered saline was purchased from Oxoid Ltd. (Hampshire, England). Methanol was purchased from ECP-Laboratory Reagent (Auckland, New Zealand). Ultrapure water was generated by a Milli-Q water purification system.

### 2.2. Preparation of Black Tea Infusion (BTI)

Bagged black tea was used to make tea infusion. One tea bag (2 g-equivalent of tea leaves) per 100 mL RO (Reverse osmosis) water was used to extract polyphenol rich infusions, with the mixture being soaked at 85 °C for 15 min in a water bath. After the extraction, the bags were removed and the solution was cooled to 40 °C in ice water for further experiments [34].

### 2.3. (Stirred) ADG Preparation

In general, tea-enriched ADG samples were made by acidification of tea-enriched reconstituted skimmed milks (T-RSM). Regarding the preparation of a T-RSM, SMP was dissolved in the water-BTI mixture solvent; the volume fraction of BTI in the water-BTI mixture solvent was manipulated for achieving the targeted proportion of BTI in the overall T-RSM system. The targeted BTI proportions in individual T-RSM samples were 15%, 30%, 45% and 60% (*w*/*w*). Also, each of the yield T-RSM sample should contain 10% (*w*/*w*) milk dry matter. Then, ADG samples was prepared directly from the T-RSM samples containing different proportions of tea infusion. RSM samples without the supplement of BTI were used to prepare the negative control ADG samples. The ADG samples were prepared following Vega and Grover [35] method with slight modification. Briefly, for instance, for each replicate milk sample, 200 g of SMP was dissolved in water-BTI mixture solvent to yield 2000 g T-RSM under vortex condition for 30 min and then refrigerated at 4 ± 1 °C overnight before further use. As aforementioned, the volume fraction of BTI in T-RSM was controlled for achieving the targeted proportion. The T-RSM was heated in water bath at 85 °C for 20 min and then rapidly cooled to 40 °C in an ice-water bath. GDL (1.3% *w*/*w*) was added and dissolved in the T-RSM system with gentle stir. Different T-RSM (including RSM control) samples were then incubated at 35 °C until pH reached pH 4.55. At this point, the set ADG samples were prepared. Twelve replicate set gels (batches) were made for each sample. Subsequently, the individual set gels were homogenized (using IKA T25 Digital Ultra-Turrax, Werke GmbH & Co. KG, Staufen, Germany, at 4500 rpm, for 1 min) [36,37] for making stirred gels. These stirred gels containing 0%, 15%, 30%, 45% and 60% (*w*/*w*) of black tea infusion were the key samples studied in this research and these samples are named as ADG_0%_ (negative control), ADG_15%_, ADG_30%_, ADG_45%_ and ADG_60%_ respectively. The stirred gels were stored at 4 ± 1 °C for further analysis. Related measurements were carried out on day 1, 7, 14, 21 and 28 of cold storage.

### 2.4. pH and Ca^2+^ Content Measurement

The pH of ADGs was determined at 20 ± 2 °C using a digital pH meter after calibration (SevenEasy pH, Mettler-toledo GmbH, Schwerzenbach, Switzerland).

The free Ca^2+^ content in ADG system was determined by a portable Calcium Ion-Selective Electrode (B-751, LAQUA Twin Calcium Ca++ Ion Meter, Horiba, Japan) according to Kosasih, et al. [38] and the instruction of the Ca^2+^ meter. The Ca^2+^ meter was calibrated by a 2-point calibration using 150 ppm and 2000 ppm Ca^2+^ standards before use and the error was below 20%. Samples stayed for 1 h at ambient temperature and temperature was measured by a thermometer right before pH and free Ca^2+^ content were measured.

pH and free Ca^2+^ content of stirred ADG samples were carried out on the day 1, day 7, day 14, day 21 and day 28 of cold storage. Measurements were carried out in triplicate.

### 2.5. Texture Characteristics

Textural Attributes of Stirred Gel. The textural properties of stirred gel were characterized by back-extrusion method described by Ciron, et al. [14]. A 5-kg load cell was used and the samples were tested in cylinder pot (50 mm internal diameter) at 15 °C, using an extrusion disc (Φ = 35 mm) operating at a set-speed of 1.0 mm/s to a 30 mm depth. Firmness and cohesiveness values were calculated from the obtained profiles using the software provided by Stable Microsystems. Textural profile of stirred ADG samples were carried out on the day 1, day 7, day 14, day 21 and day 28 of cold storage. Textural profiles were tested in triplicate for stirred gel samples.

### 2.6. Phase Stability

The water-holding capacity (WHC) and syneresis rate of ADGs were measured according to the method of Ciron et al. [32] with minor modifications using a centrifuge (Heraeus^®^ Multifuge X3R, Heraeus Co., Hanau, Germany). For measuring WHC, the stirred ADG samples (1.0 g) in 1.5-mL microtubes (Axygen^®^) were centrifuged at 15,000× *g* for 15 min at 25 °C. The WHC was expressed as % (pellet/sample, *w*/*w*). The extent of syneresis (EOS) was determined by centrifuging stirred ADG samples (1.0 g) in 1.5 mL tubes at 101× *g* for 60 min at 5 °C. The EOS was expressed as % (supernatant/sample, *w*/*w*). WHC and EOS measurements of the stirred ADGs were conducted in quadruplicate. WHC and EOS measurements of stirred ADG samples were carried out on the day 1, day 7, day 14, day 21 and day 28 of cold storage.

### 2.7. Antioxidant Capacity Evaluation

Sample Extract. The sample (1 g for both tea infusion and gel samples) was placed in a 50-mL plastic pot with 20 mL 50% methanol solution and stirred overnight (speed 3, RT 15 Power 15-position analogue hotplate stirrer, IKA, Staufen im Breisgau, Germany) at ambient temperature. Extracts were stored at −20 °C until required.

Total Phenolic Content (TPC). The total phenolic content (TPC) of the sample extracts was determined by a spectrophotometer in triplicate using Folin-Ciocalteu (F-C) reagent according to the method described in previous studies [39,40] with slight modifications. 500 µL of sample was added to the test tubes followed by 2.0 mL of sodium carbonate (7.5 g/100 mL) and 2.5 mL of 0.2 mol/L Folin-Ciocalteu reagent. The samples were mixed thoroughly and stored in the dark for 2 h before the absorbance was measured at 760 nm using VWR V-1200 Spectrophotometer (VWR International Co., Pennsylvania, USA). TPC was expressed as mg gallic acid equivalents (GAE) per 100 g of fresh material.

DPPH Assay. The DPPH radical-scavenging activity was assayed by the method reported by Al-Dabbas et al. [41]. DPPH was dissolved in methanol to get a concentration of 0.1 mM. To 500 μL of sample extract, 1 mL 0.1 mM DPPH solution and 1.5 mL of methanol were added and mixed by vortex. The absorbance was measured at 517 nm (VWR International Co., Pennsylvania, USA) after the solution was kept at room temperature for 30 min in the dark, methanol was used as the control. The DPPH radical-scavenging activity was expressed as mg Trolox equivalents (TE) per 100 g of fresh material. Each sample was analysed in triplicate.

Ferric Reducing/Antioxidant Power (FRAP) Assay. FRAP was assessed according to Khanizadeh, Tsao, Rekika, Yang and DeEll [40] with slight modifications. A fresh working solution of FRAP reagent was prepared each time by mixing acetate buffer (300 μM, pH 3.6), a solution of 10 mM TPTZ in 40 mM HCL, and 20 mM FeCl_3_•7H_2_O at 10:1:1 (*v*/*v*/*v*). 250 μL of standards of iron (II) sulphate (FeSO_4_•7H_2_O) or sample extracts were added to 2.5 mL of the FRAP reagent and the absorbance at 593 nm recorded immediately after the addition of the sample and after 2 h incubation at 37 °C [42] (p.39). The results were expressed as μmol Fe2+/g sample. Each sample was analysed in triplicate.

ABTS radical scavenging capacity. The ABTS radical scavenging assay was based on the method of Elfalleh, et al. [43]. ABTS working solution was prepared by mixing colourless ABTS stock solution (7 mM in water) with 2.45 mM potassium persulfate (1:1) and then maintain the reaction for 16 h in the dark at room temperature. Before analysis, the working solution was diluted to an absorbance of 0.70 (±0.02) at 734 nm with PBS (pH 7.4) and 3 mL of ABTS·+ transferred to a cuvette. After the addition of 300 μL Trolox or sample extract, the mixture was well mixed by blowing with pipette, allow to stand 6 min and absorbance read at 734 nm. All samples were assayed in triplicate. The results were expressed as Trolox equivalents (TE).

TPC, DPPH, FRAP and ABTS measurements of stirred ADG samples were carried out on the day 1, day 7, day 14, day 21 and day 28 of cold storage.

### 2.8. Microstructure of ADG

Scanning Electron Microscope (SEM). The bulk microstructure characterization method was developed according to Kalab [44]. Approximately 3 mm^3^ cubes of chilled stirred gel samples were coated in a thin layer of low melting point agarose (3%) before being placed into primary fixative (3% glutaraldehyde 0.1 M sodium cacodylate buffer, pH 7.2) for at least 8 hours at ambient temperature. The samples were then washed three times (10–15 min each) in sodium cacodylate buffer (0.1 M, pH 7.2) followed by post fixation in 1% osmium tetroxide (in sodium cacodylate buffer) for 1 hour at room temperature and another three buffer washes. Finally, the samples were dehydrated in gradient ethanol series (25%, 50%, 75%, 95% and 100%) for 10–15 min each before a final 100% ethanol wash for 1 h.

Samples were critical point dried using liquid CO_2_ as the CP fluid and 100% ethanol as the intermediary (Polaron E3000 series II critical point drying apparatus, Quorum Technologies, Laughton, UK). Samples were torn to expose structure, mounted on to aluminium stubs, sputter coated with approximately 200 nm of gold (Baltec SCD 050 sputter coater, Schalksmühle, Germany) and viewed in the FEI Quanta 200 Environmental Scanning Electron Microscope with the EDS Detector (EDAX Genesis, Sydney, Australia) at an accelerating voltage of 20 kV. Analyses of ADG microstructure were conducted at Massey University, Palmerston North, New Zealand [45,46,47,48].

Confocal Laser Scanning Microscopy (CLSM). The planar microstructure of the protein arrangement in stirred ADG was investigated using CLSM. The method was adapted from Ciron, Gee, Kelly and Auty [32] with modifications. A drop of stirred gel sample was placed in a concave slide and 40 μL each of 0.2 g/L Nile red (in methanol) and fast green (in water) were added before being covered with a coverslip.

Imaging was carried out using the Leica DM6000B SP5 confocal laser scanning microscope system with LAS AF software (version 2.7.3.9723; Leica Microsystems Berlin, CMS GmbH, Berlin, Germany). Images were acquired with a HCX PL APO CS 10x (N.A. 0.40), HCX PL FLUOTAR 40x (N.A. 0.75) and HCX PL APO CS 100x oil (N.A. 1.40). Nile red and fast green were sequentially imaged through excitation at 488 nm (argon laser) and 633 nm (HeNe 633 laser) (respectively) and emission collection at 498–569 nm and 643–787 nm (respectively).

### 2.9. Statistical Analysis

One-way analysis of variance (ANOVA) (with Fisher comparison) and principle component analysis (PCA – as of Figure 1) were carried out using Minitab 17.0 (Minitab, State College, PA, USA) and the significance level was set at *p* ≤ 0.05.

## 3. Results and Discussion

### 3.1. Physicochemical Characteristics

The physicochemical characteristics of ADGs are shown in Table 1. As it is shown in Table 1, plain ADG_0%_ showed the highest values in EOS in comparison with other ADG samples on day 1. Throughout the cold storage period, in general, decrease trends were observed in pH, Ca^2+^ concentration, and EOS for all ADG samples. Significant decreases of Ca^2+^ concentration were only observed in ADG_45%_ and ADG_60%_ on day 1 (in comparison to ADG_0%_). The decrease of Ca^2+^ concentration may due to the interaction between calcium ion and polyphenols or oxalate in tea. Charrier and coworkers [49] stated that the oxalate in teas may bind with calcium ion. Yamada et al. [50] used infrared spectroscopy (IR) and matrix-assisted laser desorption ionization technique with a time-of-flight mass spectrometer (MALDI-TOF-MS) found out that the calcium ion increased the amount of tea stain greatly due to the combination of phenolic compounds and calcium ions. Such finding provided an evidence that calcium-bridged polyphenols complex may be formed. Another study [51] also evidenced occurrence of EGCG-Ca^2+^-EGCG bridging effect. These Ca^2+^-phenol interaction mechanisms help explained our results regarding the observed decrease of Ca^2+^ from a molecular point of view. The slight decrease trend of pH upon increasing the incorporation amount of tea infusion (Table 1) is attributed to the relatively lower pH of black tea infusion (pH 5.18) [21].

Stirred ADGs are viscoelastic semi-solid material, poor gel structure and poor stability are associated with higher risk of shrinkage and subsequent expulsion of whey solution [32], so a relatively phase stable ADG should be able to retain a certain amount water over a storage time [52]. Interestingly, by including tea infusion in the stirred ADG, the syneresis rate tended to decrease when the stirred gel was just made (day 1, Table 1). A study demonstrated an opposite phenomenon, in which the authors found that polyphenols especially at higher concentrations are able to reduce the elasticity of acidified milk gels and induce extensive shrinkage of gel system, therefore, more syneresis [53]. Such mechanism can explain our EOS results on day 28. Larger polyphenols with more aromatic rings and hydroxyl groups have stronger affinity to casein for forming protein–phenol complex [1]. Such type of complex may result in tighter gel network then cause relatively more release of whey fraction [27]. In the same study [27], the coworkers found two types of protein–phenol complex systems when milk proteins interact with phenolic compounds from either green tea or Pu-erh tea. These two different complex systems resulted in different texture and syneresis rate. Green tea infusion was able to reduce syneresis; however, Pu-erh tea promoted syneresis in acidified dairy gel [27]. The current research showed that black tea had even stronger ability than green tea (as shown in the reference [27]) in terms of reducing the syneresis rate of ADG on day 1 (Table 1). After 28 days of storage time, ADG_0%_ and ADG_15%_ were not significantly different for EOS; however, for gels with higher levels of tea infusion inclusion the EOS were significantly increased indicating time is an important factor for developing the impact of protein–phenol complex on gel structure (Table 1). Such explanation is confirmed by (SEM) microstructure images (e.g., Figure 2e vs. Figure 3e) which will be discussed later.

As for WHC, ADG_15%_ and ADG_30%_ acquired the highest values for day 1; however, after 28 days ADG_15%_ had the highest WHC than other gel samples including the control gel without inclusion of tea infusion (*p* < 0.05), thus the relatively lower amount of tea infusion (15%) may help to maintain the gel structure of ADGs. Reports have shown that exopolysaccharides (EPSs) can act as an agent which is capable of thickening, stabilizing, emulsifying and gelling as well as water-binding in the food system and EPSs can be produced by certain strains, such as *Lactobacillus paracasei*, therefore an increase in EOS was observed through cold storage [14,52]. However, in this research, no starter culture was used and the gelation was induced by (slow) direct acidification due to the hydrolysis of GDL. Such results suggest that not only starter culture and its products but also physicochemical interactions between dairy components and tea phenolic compounds can alter gel structure and texture, therefore, the gel physical stability over a period. The same mechanism has been discussed previously elsewhere [27], although, in that research the dairy gel was induced by fermentation of starter culture, the author attributed the texture and stability features to the protein–phenol interactions.

### 3.2. Texture Characteristics

The textural characteristics of stirred ADGs are shown in Table 2. In general, firmness describes the force needed for causing a certain deformation. Cohesiveness represents for the threshold of extent of a material deformation for triggering an irreversible rupture [54]. The former parameter provides good indication about how hard or how soft a gel material, whereas, the later indicates the tenacity of a gel material (e.g., brittleness). On day 1, incorporation of tea infusion regardless the addition level resulted no change in firmness in comparison to the control (Table 2). Over the storage time (> day 1), all gel samples showed an increase in gel firmness in comparison to day 1, this is due to the reformation of gel structure after shearing the gel on day 0 and ongoing fusion of casein particles such as dissociation of casein micelles at relatively low pH and rearrangement of bonds and strands [55]. The similar phenomenon was observed in a previous work in which the authors observed significant increase of storage modulus of stirred yogurt gel over a period of storage (up to 35 days) [56]. Apparently, the current results showed that the higher addition amount of tea infusion (ADG_60%_, Table 2) retarded the fusion process of casein particles in the gel system over the 28 days storage time. In summary, including tea infusion at extremely high level (e.g., 60%) tends to make dairy gel relatively softer over storage time. The reason is not clear that why ADG_45%_ had slightly higher firmness than ADG_30%_ after 28 days of storage. The impact of tea infusion on dairy gel firmness is not fully clear and the published data is lack of consistency. Opposite observations were reported, for instance, one group found that green tea extracts were able to increase the firmness of yogurt. And the phenomenon was attributed to flavonoid–protein cross-linking [57]. However, the authors did not mention the details of the gel preparation procedure. It is important to point out that the shearing process of set gel for the preparation of stirred is a critical step, which determines the firmness of the final sample. If the gel shearing process was not consistent, consequently, it is impossible to make relevant discussion about the causality for experimental observations. Moreover, it was found that green tea extracts significantly increased hardness of cheese gel but decreased its cohesiveness and this is due to the net effect of moisture reduction and alter microstructure of cheese gel in which green tea extract was included [30]. Similar to our findings, Najgebauer-Lejko et al. [27] also found that tea infusion may be able to decrease firmness of acidified dairy gel, although green tea (rather than black tea) and lower addition levels were used (5–15%) in that research.

Unlike the situation for firmness, incorporation of tea infusion (15–60%) in ADG significantly reduced the cohesiveness on day 1 (Table 2). Such results indicated that addition of tea infusion may make the acidified dairy gel slightly more brittle. The opposite results were reported for green tea and Pu–erh tea enriched yogurt systems, in which cohesiveness was increased upon addition of tea infusion (up to 15%) [27]. However, it is important to point out that not only were different teas used in the two studies, but also relatively lower tea infusion addition levels were investigated in the mentioned reference. Najgebauer–Lejko et al. believed that the increase of cohesiveness is due to the existence of protein–phenol interactions and such interactions strengthened the food internal bonds [27]. However, such deduced mechanism does not explain the contradictory trends between firmness and cohesiveness as found in the mentioned study. In the current research, the changing trend of cohesiveness over 28 days of storage was similar to the firmness-changing trend. The change of firmness and cohesiveness upon addition of tea infusion may be attributed to the alteration of microstructure during storage period [30]. The mechanism will be elaborated later in the microstructure section.

### 3.3. Antioxidant Capacity

To determine the antioxidant abilities of the obtained stirred ADGs and tea infusion, we chose three methods which allowed us to measure both the ability to reduce pro-oxidant metal ions (FRAP assay) and radical scavenging activity (DPPH assay and ABTS assay) along with TPC measurement. The results are shown in Table 3.

Based on the results from FRAP, DPPH and ABTS radical assays, the strongest antioxidant capacity appeared in tea infusion (TPC: 1632.56 ± 9.99 GAE µg/g; FRAP: 17.20 ± 0.09 Fe^2+^ equivalent µmol/g; DPPH: 9.26 ± 0.08 TE µmol/g and ABTS: 9.30 ± 0.04 TE µmol/g) compared to dairy gel samples. The difference in antioxidant activity is due to the different supplement volume of the original tea infusion. Based on the original antioxidant capacity of black tea infusion and the proportion of tea infusion supplementation, we calculated the theoretical antioxidant capacities for tea-enriched ADGs. By comparing the theoretical and practical values, we found that the acidified dairy gels did not reduce the antioxidant activity from the tea infusion. The higher proportion of tea infusion presented in ADGs the more antioxidant capacity and TPC. Interestingly, the increasing rate in TPC was much lower than the increasing rate of tea infusion supplementation (e.g., comparison between ADG_0%_, ADG_15%_, and ADG_30%_) which indicated that the polyphenols in tea were binding with calcium ion or protein as reported by Tanizawa, et al. [58] and Spiro and Chong [59], respectively. These previous findings are in a good agreement with our results in which we observed that the free Ca^2+^ were decreased upon the increase of the addition level of tea infusion (Table 1). Surprisingly, some antioxidant parameters were remained at higher level at day 28 compared to results from day 7, 14 and 21. Although the mechanism has not yet been clear, similar results were reported by Najgebauer-Lejko, et al. [15]. The authors studied yogurt gels with incorporation of different levels of either green tea infusion or Pu-erh tea infusion, and their results showed that the antioxidant activities decreased until day 21 for both types of tea but such activity increased back to nearly the initial value on day 28.

Overall, the incorporation of black tea infusion in stirred ADG significantly enhanced the antioxidant potential of ADG as a 4 to 12-fold increase in DPPH, an 8 to 40-fold in FRAP and a 6 to 18-fold in ABTS value were observed in comparison to plain ADG_0%_. The antioxidant activities were stable over the 28 days shelf life. Moreover, such activities may be effectively increased by incorporation of increased proportion of black tea infusion.

### 3.4. Principle Component Analysis (PCA)

Plots of PCA is shown in Figure 1. PC1 and PC2 including physicochemical properties and antioxidant properties explained 79.1% of the variation with 54.9% and 24.2% for the two PCs respectively. The main difference between PC1 and PC2 is that PC1 strongly associated with TPC (PC1 0.394; PC2 0.037), DPPH (PC1 0.385; PC2 0.156), ABTS (PC1 0.395; PC2 0.191) and FRAP (PC1 0.398; PC2 0.217) but values of EOS (PC1 0.145; PC2 −0.355), pH (PC1 0.100; PC2 −0.559) and Ca^2+^ (PC1 −0.234; PC2 −0.490) were the dominant variables for PC2. Furthermore, Figure 1a. showed the sample differentiation-based storage time (shelf life); Figure 1b. showed that gels with different proportions of tea infusion are clearly separated when taking both physicochemical and antioxidant activity associated factors into account. As it is shown in the biplot figure (in the Appendix A), the antioxidant properties were mainly determined by the incorporation rate of tea infusion in stirred ADGs, but the textural and physicochemical properties were mainly influenced by storage period.

### 3.5. Microstructure

The microstructures of the stirred ADG samples are shown in Figure 2 and Figure 3 (SEM) and Figure 4 and Figure 5 (CLSM). 

SEM micrographs showed the 3D (including z-depth) organization of protein gels (stirred ADG samples). In Figure 2, comparing the morphology and the segregated structure between ADG_0%_ and ADG_15%_, the z-depth structure is more densely packed in the ADG_15%_ (Figure 2a vs. Figure 2b). Also, the later had slightly finer protein arrangement, as shown in Figure 2, ADG_0%_ had relatively more large cavities (white arrows in Figure 2a) than those in ADG_15%_ (Figure 2b). The structure of ADG_30%_ (Figure 2c) was relatively thicker and denser among other gel samples on day 1. Figure 2c showed that the sample had fine protein arrangement resulting very small pores in the gel structure. Such microstructure explains the better phase stability (WHC, Table 1) in the gel sample containing 30% BTI. Generally, ADG_15%_, ADG_30%_ and ADG_45%_ three samples had similar spongy-like interior with few air cells and highly branched-structure. Fiszman, et al. [60] found that the smooth bridge with double network structures of dairy gel seemed to be located at the inside of network of casein micelles that could maintain the aqueous phase more effectively and reduce EOS. Although the authors did not study the impact of tea infusion on dairy protein gelation, the work clearly demonstrated that smaller pore sizes within the 3D gel network resulted in lower syneresis rate. Such observation about the relation between 3D structure of gel and its phase stability is in a good agreement with our observations. For instance, including even a relatively small volume of BTI may result in relatively smaller pore sizes for acidified milk gel (ADG_15%_, Figure 2b, white arrow) than the control gel (Figure 2a, white arrow); consequently, such structure may resulted in relatively lower EOS for ADG_15%_ on day 1 in comparison with the control gel (Table 1). Although the impact of tea addition to yogurt (or ADG) on the gel texture, antioxidant activity, lactic acid bacteria and quality related characteristics have been studied elsewhere [15,22,27,61,62], few research has been done to investigate the impact of tea infusion on microstructure of acidified milk protein gel systems. However, other phenolic compounds rich materials have been incorporated in yogurt gels and the microstructure of the gels was studied. For instance, a recent research showed that 3% apple pomace in stirred yogurt resulted in more compacted protein network and larger cavities in the gel in comparison with negative control gel [63]; moreover, Pan and co-workers found that inclusion of 5% pomegranate juice powder in set yogurt resulted in denser protein gel packing and inhomogeneous size distribution pores [64,65]. These observations in literature are in good agreement with results found in this research. Some aggregated and/or lumpy protein networks (white circles in Figure 3d,e) appeared in gels with high incorporation rate of BIT (45% & 60%) after 28 days of cold storage; these structural features cannot be observed in the same samples on day 1 (white circles in Figure 2d,e). These changes in structural organization of ADGs explain the instability of ADG_45%_ and ADG_60%_ (EOS and WHC results in Table 1).

CLSM was used for characterizing laminar microstructure of ADG samples. On day 1, ADG_30%_ (Figure 4c) and ADG_45%_ (Figure 4d) had denser protein gel blocks as pointed by white arrows. By comparing between Figure 4d,e and Figure 5d,e, more and bigger pores appeared in samples containing large volume of BTI (45% and 60%) after 28 days period of storage. The results suggested that significant transformations of gel structure took place during storage in which incorporation rate of BTI is high (45% and 60%). The structure characteristics of ADG_30%_ remained nearly the same on day 1 and day 28. Such result is consistent in both SEM (Figure 2c and Figure 3c) and CLSM (Figure 4c and Figure 5c) images. The poorest gel network was found in ADG_60%_ after 28 days storage, such structure resulted in worst cohesiveness among all gel samples (Table 2, *p* < 0.05).

In general, the structural changing trend was consistent between SEM images and CLSM images. The interactions between polyphenols and dairy proteins may be responsible to the microstructure changes of acid induced milk protein gel. The impact of tea extract components on the acidified gelation process is rather complicated as they affect both micro- and macro- structures at the same time.

## 4. Conclusions

The addition of black tea infusion as a nutritional ingredient to the acidified milk gel significantly reduced EOS on day 1 at all addition levels. However, such advantage started disappearing for ADG_45%_ after 7 days of storage; after 28 days, only ADG_15%_ showed similar EOS as the negative control gel and other gels samples all had higher EOS. At 30% or higher incorporation rates of BTI, the texture of the stirred gel became relatively softer and more brittle after 28 days of cold storage. Both phase stability results and changing trend of texture can be explained by the micrographs, SEM and CLSM images provided complementary information regarding the structural characteristics of BTI enriched milk gel systems. These micrographs may be used as good references for future research about tea-enriched ADG system (e.g., yogurt), since the microstructure features of such type of gel system have not yet been extensively reported. The inclusion of BTI led to remarkable increases of the antioxidant activity for the ADG samples. Such increased antioxidant activity is attributed to the increased TPC derived from BTI. The antioxidant capacity obtained from BTI was relatively stable during a 4-week cold storage. Overall, we recommend that 15% incorporation rate of BTI in ADG is the optimum. At this rate, after 28 days storage, EOS and gel firmness was not compromised and WHC was even higher in comparison with the negative control gel. Moreover, at the end of shelf life the TPC was increased nearly 50% in ADG_15%_ compared to ADG_0%_. Both similar and contradictory results were found in literature resources regarding the impact of supplement of tea infusion on the texture of acidified milk gel. Also, detailed mechanisms of interactions between tea infusion components and dairy proteins remain unclear. Therefore, further research is still needed for revealing these mechanisms.

## Figures and Tables

**Figure 1 foods-09-00831-f001:**
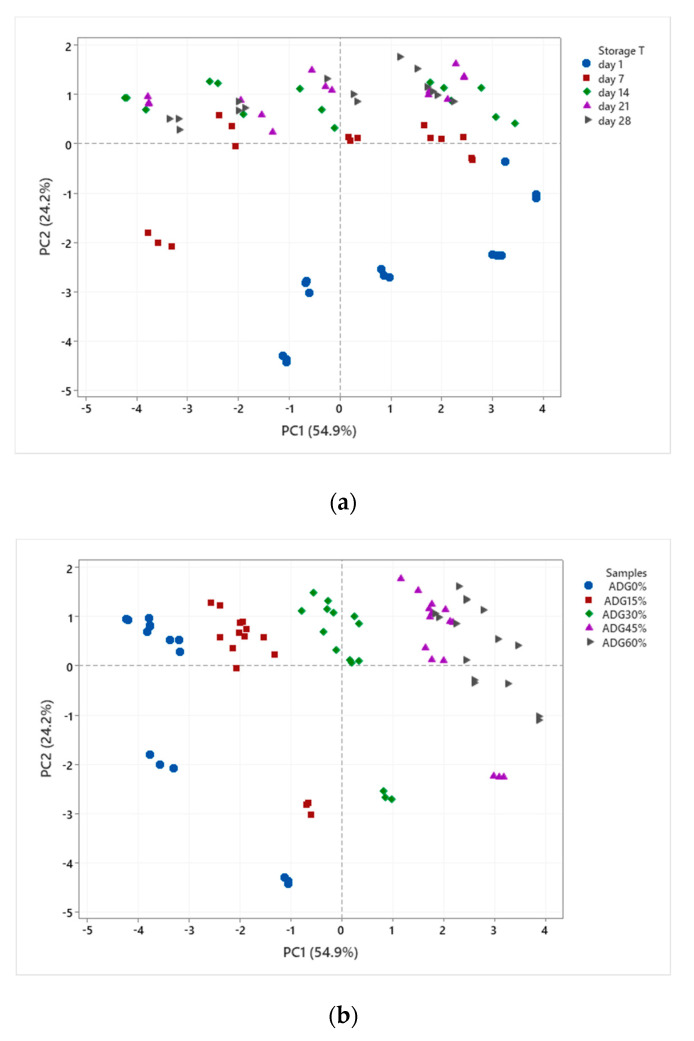
Score plots of principle component analysis (PCA). ADG_0%_: plain acidified dairy gel, ADG_15%_: acidified dairy gel containing 15% black tea infusion, ADG_30%_: acidified dairy gel containing 30% black tea infusion, ADG_45%_: acidified dairy gel containing 45% black tea infusion, ADG_60%_: acidified dairy gel containing 60% black tea infusion. PC1 strongly associated with TPC, DPPH (2,2-Diphenyl-1-picrylhydrazyl), ABTS (2,2-Diphenyl-1-picrylhydrazyl) and ferric reducing antioxidant power (FRAP); PC2 strongly associated with values of EOS, pH and Ca^2+^. (**a**) is PCA plot based on different storage time; (**b**) based on different formula.

**Figure 2 foods-09-00831-f002:**
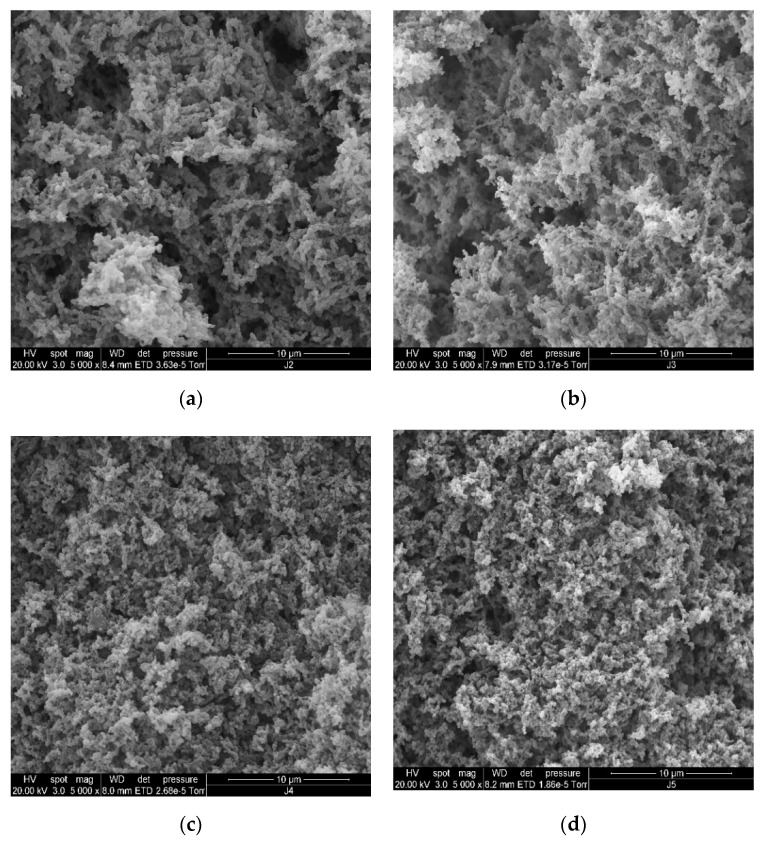
SEM images of plain ADG_0%_ (**a**), ADG with 15% black tea infusion (**b**), ADG with 30% black tea infusion (**c**), ADG with 45% black tea infusion (**d**), ADG with 60% black tea infusion (**e**) (day 1).

**Figure 3 foods-09-00831-f003:**
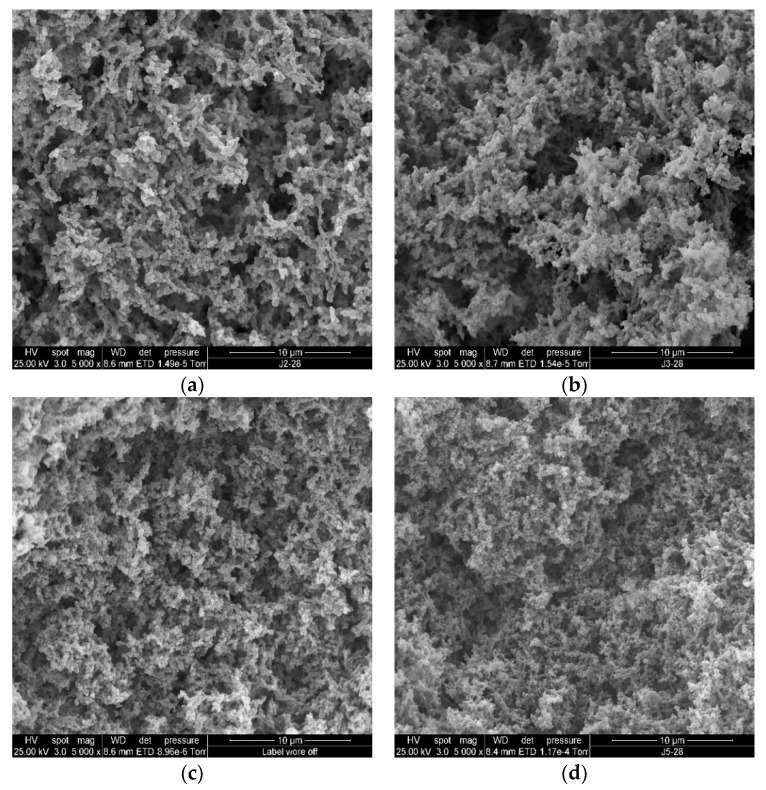
SEM images of plain ADG_0%_ (**a**), ADG with 15% black tea infusion (**b**), ADG with 30% black tea infusion (**c**), ADG with 45% black tea infusion (**d**), ADG with 60% black tea infusion (**e**) (day 28).

**Figure 4 foods-09-00831-f004:**
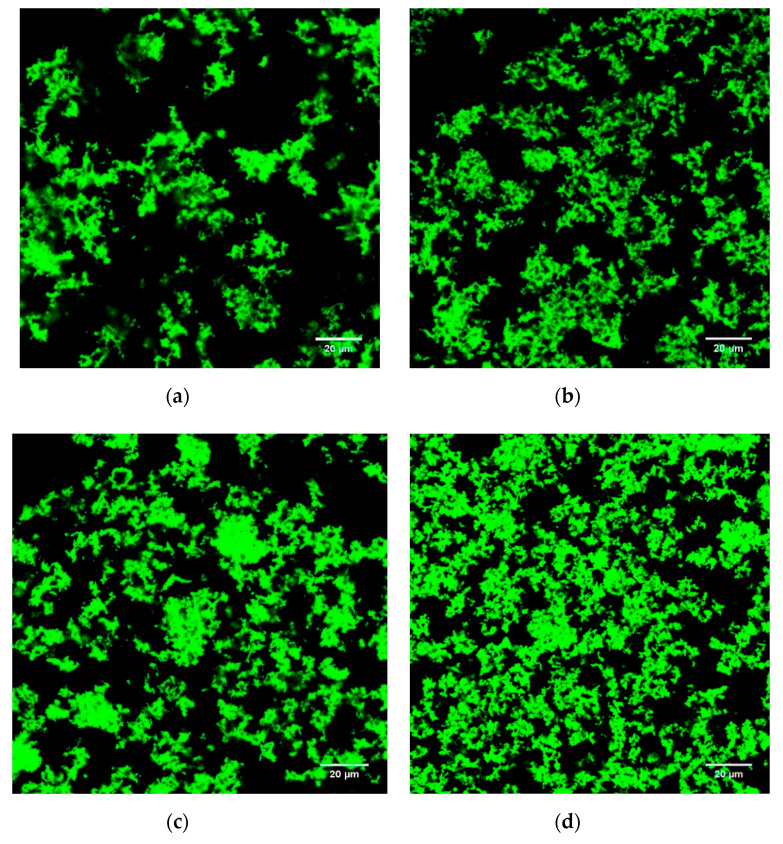
CLSM (confocal laser scanning microscopy) images of plain ADG_0%_ (**a**), ADG with 15% black tea infusion (**b**), ADG with 30% black tea infusion (**c**), ADG with 45% black tea infusion (**d**), ADG with 60% black tea infusion (**e**) (day 1). Protein stained by Fast Green FCF appears as green and non-fluorescent areas (dark areas) correspond to the serum pores.

**Figure 5 foods-09-00831-f005:**
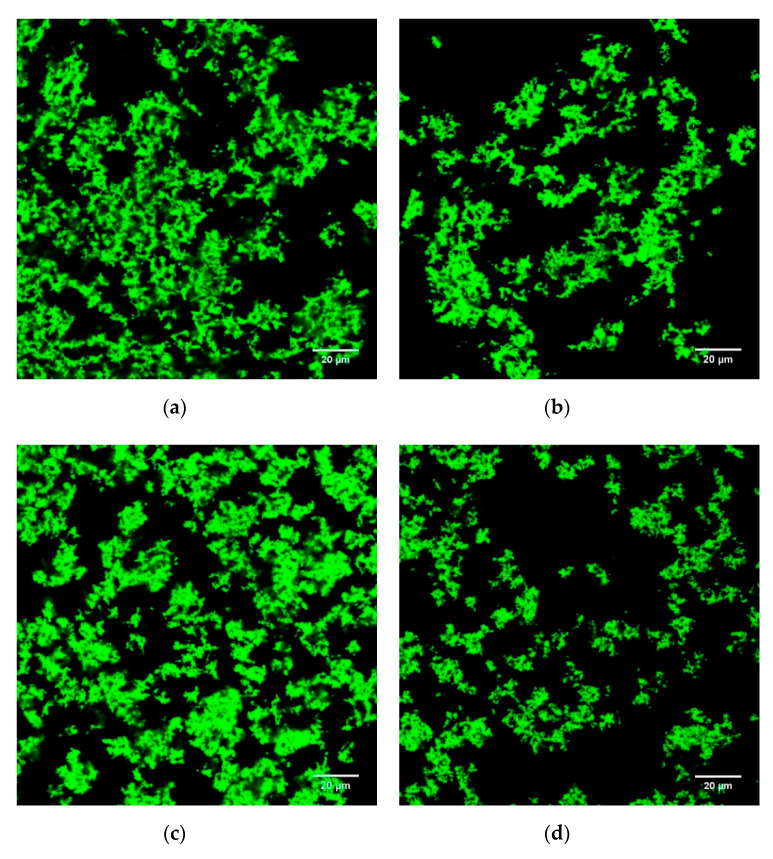
CLSM images of plain ADG_0%_ (**a**), ADG with 15% black tea infusion (**b**), ADG with 30% black tea infusion (**c**), ADG with 45% black tea infusion (**d**), ADG with 60% black tea infusion (**e**) (day 28). Protein stained by Fast Green FCF appears as green and non-fluorescent areas (dark areas) correspond to the serum pores.

**Table 1 foods-09-00831-t001:** Physicochemical characteristics between different formulations of black tea enriched acidified dairy gel (ADG) during a cold storage of 28 days (4 °C).

Physicochemical Properties	Storage Time (Days)	Gel Samples ^1^				
		ADG_0%_	ADG_15%_	ADG_30%_	ADG_45%_	ADG_60%_
pH value	1	4.55 ± 0.01 ^Aa^	4.53 ± 0.01 ^Ab^	4.54 ± 0.01 ^Aab^	4.54 ± 0.01 ^Aab^	4.53 ± 0.01 ^Ab^
	7	4.35 ± 0.01 ^Bc^	4.37 ± 0.01 ^Bb^	4.36 ± 0.01 ^Bc^	4.34 ± 0.01 ^Cd^	4.40 ± 0.01 ^Ba^
	14	4.29 ± 0.01 ^Db^	4.28 ± 0.01 ^Dbc^	4.36 ± 0.01 ^Ba^	4.36 ± 0.01 ^Ba^	4.28 ± 0.01 ^Dc^
	21	4.30 ± 0.01 ^Db^	4.33 ± 0.01 ^Ca^	4.32 ± 0.01 ^Cb^	4.28 ± 0.02 ^Dc^	4.29 ± 0.01 ^Dc^
	28	4.32 ± 0.01 ^Ca^	4.28 ± 0.02 ^Dc^	4.25 ± 0.01 ^Dd^	4.29 ± 0.01 ^Dbc^	4.31 ± 0.01 ^Cab^
Ca^2+^ concentration	1	450.00 ± 0.00 ^Ba^	443.33 ± 5.77 ^Aa^	446.67 ± 15.28 ^Aa^	353.33 ± 5.77 ^Ab^	273.33 ± 15.28 ^Bc^
	7	500.00 ± 10.00 ^Aa^	356.67 ± 11.55 ^Bb^	316.67 ± 5.77 ^Cc^	253.33 ± 5.77 ^Cd^	326.67 ± 15.28 ^Ab^
	14	356.67 ± 5.77 ^Ca^	323.33 ± 5.77 ^Cb^	326.67 ± 5.77 ^Bb^	253.33 ± 5.77 ^Cc^	263.33 ± 5.77 ^Cc^
	21	370.00 ± 10.00 ^Ca^	326.67 ± 15.28 ^Bb^	293.33 ± 5.77 ^Dc^	266.67 ± 5.78 ^Bd^	246.67 ± 11.55 ^Ce^
	28	330.00 ± 10.00 ^Dab^	336.67 ± 5.77 ^Ba^	316.67 ± 5.77 ^Cb^	246.67 ± 5.77 ^Cd^	290.00 ± 10.00 ^Bc^
EOS^2^/%	1	39.95 ± 0.89 ^Aa^	24.05 ± 0.78 ^Ad^	29.48 ± 0.73 ^Bc^	35.09 ± 1.15 ^Bb^	29.01 ± 1.10 ^Ac^
	7	37.74 ± 1.21 ^Ba^	22.99 ± 0.52 ^Ad^	31.16 ± 0.90 ^Ab^	39.02 ± 1.22 ^Aa^	27.50 ± 0.80 ^Bc^
	14	21.28 ± 0.80 ^Dc^	23.06 ± 1.23 ^Ab^	24.83 ± 0.20 ^Db^	20.34 ± 0.59 ^Ec^	28.42 ± 0.52 ^Aa^
	21	17.21 ± 0.39 ^Ee^	22.10 ± 0.92 ^Bd^	24.68 ± 0.68 ^Dc^	28.92 ± 0.60 ^Da^	26.36 ± 0.78 ^Bb^
	28	23.37 ± 0.33 ^Cd^	23.30 ± 0.96 ^Ad^	28.25 ± 0.33 ^Cb^	30.20 ± 0.27 ^Ca^	26.87 ± 0.14 ^Bc^
WHC^3^/%	1	17.13 ± 0.81 ^Eb^	19.56 ± 1.04 ^Ba^	21.10 ± 0.63 ^Ba^	15.63 ± 0.12 ^Dc^	15.30 ± 0.48 ^Cc^
	7	24.33 ± 0.69 ^Ba^	22.69 ± 0.23 ^Ab^	23.50 ± 0.78 ^Aa^	20.49 ± 0.79 ^Ac^	21.01 ± 0.65 ^Ac^
	14	27.31 ± 0.72 ^Aa^	21.93 ± 0.65 ^Ab^	20.69 ± 0.34 ^Cc^	16.34 ± 0.15 ^Cd^	15.22 ± 0.63 ^Ce^
	21	18.82 ± 0.25 ^Db^	16.84 ± 1.16 ^Cc^	23.89 ± 0.58 ^Aa^	18.07 ± 0.51 ^Bc^	18.54 ± 0.94 ^Bc^
	28	20.15 ± 0.55 ^Cb^	22.20 ± 0.32 ^Aa^	19.42 ± 0.37 ^Cb^	19.82 ± 0.32 ^Ab^	19.37 ± 0.84 ^Bb^

^A–E^ Means ± SD within a column with different superscripts differ (*p* ≤ 0.05). ^a–e^ Means ± SD within a row with different superscripts differ (*p* ≤ 0.05). ^1^ Formulation: ADG_0%_: acidified dairy gel without supplementation of tea infusion; ADG_15%_: acidified dairy gel containing 15% (*w*/*w*) black tea infusion; ADG_30%_: acidified dairy gel containing 30% (*w*/*w*) black tea infusion; ADG_45%_: acidified dairy gel containing 45% (*w*/*w*) black tea infusion; ADG_60%_: acidified dairy gel containing 60% (*w*/*w*) black tea infusion; ^2^ EOS: Extend of syneresis; ^3^ WHC: Water holding capacity.

**Table 2 foods-09-00831-t002:** Texture characteristics of stirred gel samples.

Texture Properties	Storage Time (Days)	Gel Samples ^1^				
		ADG_0%_	ADG_15%_	ADG_30%_	ADG_45%_	ADG_60%_
Firmness, g	1	14.77 ± 0.46 ^Ba^	14.73 ± 0.35 ^Ba^	14.57 ± 0.31 ^Ba^	14.07 ± 0.42 ^Ba^	14.70 ± 1.04 ^Aa^
	7	20.10 ± 0.90 ^Aa^	20.43 ± 1.25 ^Aa^	17.13 ± 0.72 ^Ab^	16.40 ± 0.72 ^Ab^	15.17 ± 1.08 ^Ab^
	14	21.13 ± 0.66 ^Aa^	20.07 ± 0.86 ^Aa^	18.60 ± 0.87 ^Ab^	16.37 ± 0.83 ^Bb^	15.33 ± 1.60 ^Ab^
	21	21.93 ± 0.55 ^Aa^	19.83 ± 1.15 ^Aa^	17.70 ± 0.27 ^Ab^	16.30 ± 0.87 ^Bb^	16.73 ± 0.15 ^Ab^
	28	20.53 ± 0.32 ^Aa^	18.80 ± 0.82 ^Aa^	18.20 ± 0.87^Ab^	19.03 ± 1.61 ^Aa^	16.57 ± 0.85 ^Ab^
Cohesiveness, g	1	−9.50 ± 0.27 ^Ca^	−9.07 ± 0.06 ^Bb^	−8.67 ± 0.40 ^Bb^	−8.53 ± 0.23 ^Bb^	−9.40 ± 0.61 ^Bb^
	7	−12.43 ± 0.50 ^Ba^	−12.63 ± 1.01 ^Aa^	−10.93 ± 0.35 ^Ab^	−10.53 ± 0.67 ^Ac^	−9.37 ± 0.31 ^Bc^
	14	−13.27 ± 0.42 ^Ba^	−12.83 ± 1.24 ^Aa^	−12.47 ± 1.25 ^Aa^	−10.27 ± 0.40 ^Ab^	−8.83 ± 0.71 ^Bb^
	21	−14.13 ± 0.51 ^Aa^	−12.80 ± 0.27 ^Aa^	−12.33 ± 1.10 ^Ab^	−10.07 ± 0.59 ^Bb^	−10.13 ± 0.06 ^Bb^
	28	−13.73 ± 0.12 ^Aa^	−12.57 ± 0.15 ^Ab^	−11.80 ± 1.21 ^Ac^	−10.43 ± 0.67 ^Ac^	−10.43 ± 0.40 ^Bc^

^A–E^ Means ± SD within a column with different superscripts differ (*p* ≤ 0.05). ^a–e^ Means ± SD within a row with different superscripts differ (*p* ≤ 0.05). ^1^ Gel samples: ADG_0%_: acidified dairy gel without supplementation of tea infusion; ADG_15%_: acidified dairy gel containing 15% (*w*/*w*) black tea infusion; ADG_30%_: acidified dairy gel containing 30% (*w*/*w*) black tea infusion; ADG_45%_: acidified dairy gel containing 45% (*w*/*w*) black tea infusion; ADG_60%_: acidified dairy gel containing 60% (*w*/*w*) black tea infusion.

**Table 3 foods-09-00831-t003:** Total phenolic content and antioxidant capacity of stirred ADGs.

Physicochemical Properties	Storage Time (Days)	Gel Samples				
		ADG_0%_	ADG_15%_	ADG_30%_	ADG_45%_	ADG_60%_
TPC (GAE µg/g)	1	151.42 ± 1.94 ^Ae^	229.75 ± 6.71 ^Ad^	257.90 ± 11.66 ^Ac^	454.98 ± 17.23 ^Ab^	529.28 ± 8.92 ^Aa^
	7	117.22 ± 1.71 ^Cd^	224.08 ± 9.85 ^Ac^	228.66 ± 10.26 ^Cc^	305.48 ± 8.09 ^Cb^	503.80 ± 7.38 ^Ba^
	14	119.30 ± 0.99 ^Ce^	159.08 ± 6.19 ^Dd^	211.79 ± 8.20 ^Dc^	348.37 ± 7.88 ^Bb^	441.32 ± 10.71 ^Ca^
	21	121.41 ± 6.44 ^Ce^	208.55 ± 9.31 ^Bd^	236.00±4.24 ^Bc^	303.85 ± 6.03 ^Cb^	422.99 ± 2.60 ^Da^
	28	129.25 ± 7.14 ^Be^	194.81 ± 3.34 ^Cd^	244.67 ± 5.91 ^Bc^	301.10 ± 6.03 ^Cb^	388.34 ± 5.99 ^Ea^
DPPH (TE µmol/g)	1	0.94 ± 0.06 ^Ad^	1.35 ± 0.07 ^Ac^	4.58 ± 0.29 ^Ab^	4.77 ± 0.07 ^Ab^	5.17 ± 0.06 ^Ba^
	7	0.86 ± 0.02 ^Ae^	1.27 ± 0.03 ^Bd^	4.53 ± 0.04 ^Ac^	5.02 ± 0.13 ^Ab^	5.17 ± 0.01 ^Ba^
	14	0.98 ± 0.04 ^Ae^	1.26 ± 0.07 ^Bd^	4.33 ± 0.14 ^Ac^	4.89 ± 0.08 ^Ab^	5.14 ± 0.06 ^Ba^
	21	0.93 ± 0.08 ^Ad^	1.18 ± 0.01 ^Bd^	4.48 ± 0.15 ^Ac^	4.85 ± 0.26 ^Ab^	5.24 ± 0.11 ^Aa^
	28	0.97 ± 0.13 ^Ae^	1.29 ± 0.09 ^Bd^	4.56 ± 0.15 ^Ac^	5.01 ± 0.07 ^Ab^	5.36 ± 0.08 ^Aa^
FRAP (Fe^2+^ equivalent µmol/g)	1	0.63 ± 0.03 ^Ce^	2.74 ± 0.12 ^Cd^	7.17 ± 0.13 ^Dc^	11.56 ± 0.28 ^Cb^	13.62 ± 0.23 ^Ba^
	7	0.72 ± 0.03 ^Be^	4.42 ± 0.18 ^Bd^	8.73 ± 0.28 ^Ac^	11.73 ± 0.13 ^Cb^	12.65 ± 0.05 ^Ca^
	14	0.67 ± 0.02 ^Be^	4.53 ± 0.06 ^Bd^	7.82 ± 0.2 ^Cc^	13.01 ± 0.23 ^Ab^	13.54 ± 0.11 ^Ba^
	21	0.81 ± 0.03 ^Ae^	5.17 ± 0.09 ^Ad^	8.29 ± 0.12 ^Bc^	12.48 ± 0.07 ^Bb^	14.43 ± 0.19 ^Aa^
	28	0.80 ± 0.03 ^Ae^	5.24 ± 0.13 ^Ad^	8.66 ± 0.18 ^Ac^	12.62 ± 0.08 ^Bb^	12.94 ± 0.19 ^Ca^
ABTS (TE µmol/g)	1	0.30 ± 0.05 ^Ae^	1.78 ± 0.03 ^Bd^	3.03 ± 0.10 ^Ac^	3.68 ± 0.03 ^Bb^	4.34 ± 0.05 ^Aa^
	7	0.24 ± 0.03 ^Ae^	1.86 ± 0.04 ^Ad^	3.09 ± 0.04 ^Ac^	3.70 ± 0.16 ^Bb^	4.09 ± 0.11 ^Ba^
	14	0.31 ± 0.05 ^Ae^	1.88 ± 0.02 ^Ad^	3.07 ± 0.07 ^Ac^	3.84 ± 0.06 ^Bb^	4.15 ± 0.07 ^Ba^
	21	0.31 ± 0.06 ^Ae^	1.90 ± 0.02 ^Ad^	3.08 ± 0.05 ^Ac^	4.15 ± 0.12 ^Aa^	3.99 ± 0.13 ^Bb^
	28	0.29 ± 0.03 ^Ae^	1.86 ± 0.04 ^Ad^	3.11 ± 0.04 ^Ac^	4.21 ± 0.05 ^Aa^	4.07 ± 0.12 ^Bb^

^A–E^ Means ± SD within a column with different superscripts differ (*p* ≤ 0.05). ^a–e^ Means ± SD within a row with different superscripts differ (*p* ≤ 0.05). ^1^ Formulation: ADG_0%_: acidified dairy gel; ADG_15%_: acidified dairy gel containing 15% (*w*/*w*) black tea infusion; ADG_30%_: acidified dairy gel containing 30% (*w*/*w*) black tea infusion; ADG_45%_: acidified dairy gel containing 45% (*w*/*w*) black tea infusion; ADG_60%_: acidified dairy gel containing 60% (*w*/*w*) black tea infusion; EOS: Extend of syneresis; WHC: Water holding capacity.

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
