# Peer review of "Effect of Black Tea Infusion on Physicochemical Properties, Antioxidant Capacity and Microstructure of Acidified Dairy Gel during Cold Storage"

_foods, 2020, doi:10.3390/foods9060831_

Round 1

Reviewer 1 Report

The present data deals with the effect of black tea infusion on physicochemical properties and microstructure of acidified dairy gel. I believe that the topic its interesting, but the manuscript presents serious flaws.

Relevant research is not cited in the introduction, materials and methods are not clear enough, there are acronyms along the manuscript that are not identified anywhere (and makes difficult to understand what the authors are referring to), a strong discussion is missing (many times authors just read the data), I am not a native speaker, but I noticed that there are many grammar mistakes (writing should be deeply reviewed), and also some basic mistakes that give a bad impression (lack use of capital letters when required, lack of space between numbers and units or lack of use of superscript/ Subscript among all the paper, uneven naming of Figures…).

Authors should make big changes and improve significantly all the manuscript for final publication. 

Introduction

General comment: Previous articles that worked with tea/ tea extracts inclusion un dairy gels/yogurts are not reported in the introduction. These references must be included.

Line 49-51. The following sentence needs a reference:

As antioxidants play important roles in preventing or inhibiting oxidation of cellular components, adequate intake of these compounds is beneficial to protect cells from oxidative damages

Line 53-57. I think the sentence is too long and that the given dates are not necessary data for the research. I advise to make it shorter (I believe it would look better)

Line 60-62. The following sentence needs a reference:

Acidified dairy gel (ADG) produced by glucono-δ-lactone (GDL) is very common in research which has similar physicochemical properties with yoghurt but easier to make, more stable and eliminate the confusion of microbes.

Line 62 and line 64. Instead of microbes, better use culture strains.

Line 67. Maybe after is better than through.

Line 68-70. Move the sentence to the end of the previous paragraph.

Materials and methods

Line 82. Origin of Acetonitrile and Acetic acid is missing. Please add.

Line 87. During extraction, was stirring applied? Please clarify.

Line 95-96. “The proportion of tea infusion in solvent was altered for adjusting the content of tea extracts in tea-milk mixture” I do not understand what the authors mean here, that this mean that different amount of tea was added to the water during extraction? If so, it would contradict that it was stated before…please clarify this.

Line 96. How was milk reconstituted? (how long was let to hydrate before the heating). Please clarify.

Line 99. “different levels of tea extracts”, again, I am confused about the concentration of the used extracts. From what is stated before, I assume that the concentration of the extract itself is always the same, and what changes is the amount of extract added to the reconstituted milk, and therefore the concentration of the soluble compounds present in the extract in the final milk-tea mixture…. however these words make me think that I may be wrong….please clarify this.

Line 99. Please clarify how were the containers were the tea-milk systems were incubated.

Line 101. Why day 0? This makes no sense to me, as set yogurts are never consumed the same day of production…I think this measurement should be performed at least 1 day after elaboration, and followed in time. From what is stated in the article, this was not done, so I would advise to remove this data as it doesn’t bring much.

Line 102. How was the homogenization preformed? (time, speed and used equipment is missing). Please clarify.

Line 104. From what its stated before, I assumed that pH was followed during acidification process. Please make this data available (either in the manuscript or as supplementary material)

Line 105. When was pH ADGs measured?

Line 107. When was free Ca2+ measured?

Line 113. How was temperature controlled? Maybe authors measured at ambient temperature after a certain waiting time of the gel outside the fridge? Please clarify.

Please, express the measured parameters using SI Units.

Line 114. Add space between 5 and s.

Line 122. Substitute dia. for diameter. Again, how was the temperature controlled?

Line 123. Add space between 30 and mm, delete de –

Line 124. Please clarify here when these measurements are made (both for set and stirred gels), also how many batches and how many measurements per batch.

Line 125. How and when were the samples placed in the tubes? Please clarify here when these measurements are made (both for set and stirred gels), also how many batches and how many measurements per batch.

Line 137. Please clarify for all the measurements of 2.7. when the measurements were made (days), how many batches and how many measurements per batch. It is not clear if measurements are made to the gels or to the tea extracts. I assume that is made to the gels, but this needs to be clarified

Line 140. Add an and between the two citations.

Line 170. Add a space between 20 and min. Please check all the units, number and units should always be separated by a space. The only case where this doesn’t apply is for %.

Line 179. Please clarify here when this measurement was performed, also how many batches and how many measurements per batch.

Line 182. Use superscript when needed. Apply to all the paper.

Line 197. CLSM was made to the stirred gels? Please clarify.

Discussion

Line 228. The reading of the results is not correct. Ca2+ concentration decreases on day one for ADG15% ADG30% and ADG45%

Line 231. Change Yamada [27] and colleagues for Yamada et al. [27].

Line 236. Change A study of by the study of.

Line 236-238. I do not understand the need of this statement, unless you correlate it to the observations of the microscopy pictures of this papers.

Line 242. I advise to add a certain amount of before water.

Line 242. Gained does not sound right. I advise to say All ADGs presented lower EOS values. Anyways, lower than what? Day 1 is the first data point…. On top of that, results are not discussed. If you look to the data when looked individually, all the gels (with the exception of ADG15% decreased the syneresis when comparing day 28 with day 1. If you know compare the synereis values of the different gels, at day 28 ADG and ADG15% showed the lowest syneresis values, but with no significant differences among that. Authors should elaborate a discussion on that.

Line 244. Change higher for the highest

Line 245. Higher than what? Make clear that is higher than ADG. Again, use highest.

Line 248-251. In this research no bacteria are used, so the control has no lactobacillus that could generate the mentioned EPS….I think this references can be removed, and authors should find a reference that supports why the control gel shows a low syneresis.

Line 252. As stated before, I believe that the data of set gels at day 0 brings no useful information and therefore should be removed from the article. As for stirred gels, no discussion is available at all. It seems clear to me a decrease in firmness for all the days for ADG30% and samples with higher tea concentration. I also see differences in cohesiveness. Please elaborate.

Line 272. Always write the first letter of Tables and Figures with capital letter. Apply to all the manuscript. Also, data are presented in duplicate (Figure and Table give the same data), please choose one way to show the data but do no present things in duplicate.

Figure 1. What is BT?

Line 292-293. I don’t understand this sentence. I guess BT is the tea extracts and authors want to say that it showed a higher antioxidant capacity than the gels?

Line 294. Would obtain? Shouldn’t it be obtained?

Line 294-295. “…and there were significant differences between different formulation”. Please check the writing style and rewrite.

Line 295-297. Here authors should also correlate with their own Ca2+ results previously reported.

Line 299. Any hypothesis of why this is happening? What Najgebauer-Lejko, et al. [35] say in their paper?

Line 301. Figure not figure…(as a lot of times in the manuscript)

Line 304. Please check the writing style and rewrite.

Line 345-347. Its the other way around.

Figure 3 a. Can authors explain what information is this graph bringing? It is not possible to know the tea extract concentration of each point. The authors state (line 348-349) that from this figure it can be concluded that the textural and physicochemical properties were mainly influenced by storage period…in which sense? According to Tables 1 and 2 % of tea extract also have an influence on that….

General coment of PCA. I honestly believe that PCA is not bringing new information to the article, plus the discussion is inexistent (authors only read the data).

Figure 2. I guess this is a matter of preference, but for me is confusing that the letters are below the graphs, I would advise authors to put them at the left top of the images. Also, indicate properly to what corresponds each letter…saying form a to f is xxxxxxxxxxxxx makes unclear which graph corresponds with which sample. Was ADG45% sample not measured? Use Subscript when needed.

Table 4 and 5. Please clarify the meaning of the components.

Line 315-319. Please make clear that you are referring to the standard.

Line 324. I think is better to use decrease than decline.

Line 325. With respect to BTI?

Line 326. Protein peak increase is related to the presence of milk. Please rewrite, as presented it looks like the authors want to say that is an effect of the decrease of polyphenols in all ADGs.

Line 326. When compare? Is not when comparing? Please check carefully all the English…there are some obvious mistakes.

Line 331. Figure not figure….

Line 351. Please, use uniformly Figure or Fig. along all the manuscript. Also, Figure numbers are wrong Figures 4 and 5 correspond to SEM pictures and Figures 6 and 7 to CLSM. Please correct.

Figures 4, 5, 6 and 7. Remove numbers next to letters (e.g. instead of a-1 use a), is confusing. Also, I advise to put these letters on the left top of the images. Make scale more visible (you can use Photoshop or GIMP(free) for that). Please use arrows to show in which points of the pictures the authors see what they state in the discussion.

Line 359. ….and the pores been smaller. Please rewrite.

Line 361. Change better for the observed trend in WHC and EOS. What its TI? Explain

Line 362. There are 5 samples, what do the authors want to say when they state all the tree samples?

Line 363-365. Agrees with the results of which sample? What did this study investigate? Is the effect of gelatin similar to the effect of tea extracts? Are there evidences of that? To make this statement authors have to elaborate more.

Line 371-372. I do not see the mentioned arrows.

Line 374. Table not table…, fig.3 fig.4….as stated before, make uniform the use of Figure of Fig. and use capital letters when needed!!!!

Line 381. There is research about the effect of tea powder/tea extracts on dairy products ( I did a quick search and some tittles popped up). Still, when discussing the microstructure only one article was cited, and worked with the addition of gelatin….I strongly advice the authors to improve the discussion of the microstructure with more relevant references.

Conclusions

Line 383-384. I find this sentence more of the introduction than of the conclusions, nevertheless, there is already some research involving the effect of polyphenol in yoghurt-like gels.

Line 388-389. I don’t think “enhancement the stability” can be stated, as it’s not true, what it can be said is that the quality of the obtained stirred gels with those tea % did not differed much from the control.

Line 389. What its TI? Explain

Author Response

We appreciate your patient reviews and and made changes accordingly. detailed response please see the attachment.

Reviewer 2 Report

General feedback after reading this manuscript:

  • After reading the manuscript, I don't know what the subject of the study; milk with tea infusion or tea infusion with milk was.
  • The Methodology section is highly questionable (more details below)
  • After reading the manuscript, I didn't find out anything new. It is known that adding an ingredient with antioxidant properties to the product will increase its antioxidant potential. It can be assumed that the increase will be proportional to the additive used.
  • Both the title, objectives and conclusions should be compatible.

Specific comments:

The admission must be rewritten. The introduction (line 40-64) should briefly describe the current state of knowledge on the subject. In the publication Line, 40-52 is a description of antioxidants, and line 53-57 is a description of tea, line 58-60 is a description of yoghurt; line 60-64 is a description of GDL. In the introduction, there is no information about factors influencing physicochemical properties and microstructure of acidified gels.

Line 65-70: how do the authors want to determine the interaction between tea infusion and dairy ingredient? By determining the antioxidant potential of an acidified gel?

Line 73: does skim milk powder contain only fat and protein? There is no information about carbohydrates, salts or ash.

Line: 90-103: The text is difficult to understand. After reading it, I don't know what was tested. Milk with tea infusion or tea infusion with milk? What is the DM of the product? The chemical composition of the product is necessary for this manuscript. The authors wrote that the final content of milk solid is 10% w/w. However, this information does not help to understand what has been investigated.

Line 96: "Reconstituted skimmed milk was heated" and where's the tea infusion? Was tea infusion only added after GDL? There is no answer to the questions: how long it was mixed.

Line 98: please explain the abbreviation BTI.

Line 100: 60% of what?

Line 101: why was the TPA test performed the same day? Because the gel structure is formed for about 24 hours. The analysis should be performed at least the next day.

Line 102: "The formed set gels were homogenized for making stirred gel" - homogenization is not the same as stirring. The parameters of the homogenization process are missing. Why did the authors mark the microstructure of the product after homogenization? All the more because they describe interactions between the ingredients?

Line 105-108: What kind of calibration solutions were used. The method of Ca ion determination was validated? If so, how did the validation process go?

Line 116-119: Hardness and Firmness are used alternatively and mean the same. It suggests standardizing the record. Why did the authors not use only the BE test in evaluating the gel texture? In the context of these tests, the determination of viscosity and G modules are more appropriate.

Line 126-133: Why such a small amount of sample (1g) was used in the WHC and EOS test? In my experience, I know that such results have a big mistake. What is the point of carrying out the EOS test on samples after homogenization?

Line 135: which sample was taken by the authors for testing. Before or after homogenization? What is the impact of the matrix (milk) on the test results?

Line 170: Please write down which filters (NY, RC …) were used. What was the analysis time for one sample?

Line 194-195: Write down which detector was used.

Line 199: "small amount of sample" is not precise. The results are pure speculation. The structure of the sample was twice destroyed; 1st during homogenization and 2nd after covering by cover slide. The sample was not rinsed. Was an autofluorescence test performed on the sample before right measurement?

Line 208-210: Why the samples were compared with the t-student test? Which ANOVA tests were used?

Table 1: Table is not readable; please insert horizontal separation lines. The results of the statistical tests of the samples between the variants should be verified again. Is the sample (4.55±0.01) significantly different from (4.53±0.01)?

Line 227: in the data in Table 1, the EOS value decreases during storage not increase

Line 241: "weak gels" and "good-quality" is not very precise.

Line 243-253: Please correct the discussion. What effect of EPS and bacteria if GDL was used in the study. Please describe the effect of the tea used?

Line 257: Table 2: Please correct the title of the table; the table is not very clear. What does "Se" mean before "Hardness"?

Line 263: "little significant" - what the authors mean? In general, the description and discussion of point 3.2 should be either corrected or deleted.

Line 273: table 3. As before, please insert horizontal lines. Why do the authors present the same data on the chart and in the table?

Line 296: "the increasing rate in TPC was much lower than the content of tea infusion which indicated the polyphenols in tea were binding with calcium ion" was the subject of the study tea extract or milk gel?

Line 300-302 and 345-349: obvious data.

Line 309: Please correct descriptions of Fig 2 and chromatograms. In their present form, they are not very readable. Where is peak 6?

Table 4 and 5 and line 315-322 these elements should be in the M&M section or should be deleted.

Line 323-329: And where is the description of results and discussion?

Line 351: stirred or homogenized. Please decide.

Line 356-381: The description should start with the information that homogenized gels were the subject of the study.

Line 359: "pores have been smaller" - how did the authors measure it?

Author Response

(The authors gave the same response as above.)

Round 2

Reviewer 1 Report

The authors improved the article. 

Some comments follow:

I strongly recommend to review writing style. For example, in lines 63-64 authors wrote this sentence: “During refrigeration storage condition, antioxidant activity for tea enriched yogurt is stable and it remained constant as shown in a study [2].” I think authors should mention the names of the researcher of this article (e.g. According to AUTHORS LAST NAMES [2] During refrigeration storage condition, antioxidant activity for tea enriched yogurt is stable and it remained constant).

For phase stability, authors stated in one of the answers to my comments that samples were transferred in tubes on the testing day using spatula. This should be more clearly stated in the phase stability materials and methods part, as can be confusing for the reader (in line 179 change “placing stirred ADG samples (1.0 g) in 1.5 mL tubes” for “placing (the day of measuring) stirred ADG samples (1.0 g) in 1.5 mL tubes”). Nevertheless, I find this odd, as the transferring process can affect the water holding capacity of the samples, and therefore, samples should have been stored in the 1.5ml tubes just after elaboration. This way of processing the samples makes the results unreliable.

For texture characteristics, authors do not mention when was the samples are transferred to the containers where measurements are conducted. It was the same day of measurement?

Through the text, clarify when mentioned decrease/increase of a parameter is respect the control (e.g. in line 293 is not mentioned, among others)

The authors improved the article. Nevertheless, it was hard to follow, as they deleted some parts with no track of changes.

Some comments follow:

I strongly recommend to review writing style. For example, in lines 63-64 authors wrote this sentence: “During refrigeration storage condition, antioxidant activity for tea enriched yogurt is stable and it remained constant as shown in a study [2].” I think authors should mention the names of the researcher of this article (e.g. According to AUTHORS LAST NAMES [2] During refrigeration storage condition, antioxidant activity for tea enriched yogurt is stable and it remained constant).

For phase stability, authors stated in one of the answers to my comments that samples were transferred in tubes on the testing day using spatula. This should be more clearly stated in the phase stability materials and methods part, as can be confusing for the reader (in line 179 change “placing stirred ADG samples (1.0 g) in 1.5 mL tubes” for “placing (the day of measuring) stirred ADG samples (1.0 g) in 1.5 mL tubes”). Nevertheless, I find this odd, as the transferring process can affect the water holding capacity of the samples, and therefore, samples should have been stored in the 1.5ml tubes just after elaboration.

For texture characteristics, authors do not mention when was the samples are transferred to the containers where measurements are conducted. It was the same day of measurement? or the day gels where elaborated? Please clarify  

Through the text, clarify when mentioned decrease/increase of a parameter is respect the control (e.g. in line 293 is not mentioned, among others)

Author Response

Thank you for your kind comments and help

We have addressed your points as below and feel that the manuscript has been greatly improved.

: “During refrigeration storage condition, antioxidant activity for tea enriched yogurt is stable and it remained constant as shown in a study [2].” I think authors should mention the names of the researcher of this article (e.g. According to AUTHORS LAST NAMES [2] During refrigeration storage condition, antioxidant activity for tea enriched yogurt is stable and it remained constant).

WE HAVE TRIED TO REWITE SUBSTANTIAL PARTS OF THE PAPER ON REFLECTION OF THIS COMMENT HOWEVER WE HAVE RETAINED THE NUMBERING ASPECT OF REFERENCING IN SOME AREAS

For phase stability, authors stated in one of the answers to my comments that samples were transferred in tubes on the testing day using spatula. This should be more clearly stated in the phase stability materials and methods part, as can be confusing for the reader (in line 179 change “placing stirred ADG samples (1.0 g) in 1.5 mL tubes” for “placing (the day of measuring) stirred ADG samples (1.0 g) in 1.5 mL tubes”). Nevertheless, I find this odd, as the transferring process can affect the water holding capacity of the samples, and therefore, samples should have been stored in the 1.5ml tubes just after elaboration. This way of processing the samples makes the results unreliable.

THANK YOU FOR YOUR VALUABLE COMMENTS. WE HAVE TRIED TO REVISE AREAS IN THE PAPER, HOWEVER WE ALSO NOTE YOUR COMMENTS ABOUT METHODOLOGY. WE ARE NOT ABLE TO REPEAT THE ANALYSIS USING THE METHODOLOGY SUGGESTED, BUT WILL TAKE HEAD OF YOUR COMMENTS REGARDING RELIABLE LABORATORY PRACTICES.

For texture characteristics, authors do not mention when was the samples are transferred to the containers where measurements are conducted. It was the same day of measurement?

APOLOGIES, WE HAVE TRIED TO ADD THIS DATA IN

Through the text, clarify when mentioned decrease/increase of a parameter is respect the control (e.g. in line 293 is not mentioned, among others)

THANK YOU, WE HAVE TRIED TO ATTEND TO THIS MATTER IN THE TRACK CHANGES OF THE DOCUMENT

The authors improved the article. Nevertheless, it was hard to follow, as they deleted some parts with no track of changes.

WE HAVE INCLUDED TRACK CHANGES ON ALL ALTERATIONS

Some comments follow:

I strongly recommend to review writing style. For example, in lines 63-64 authors wrote this sentence: “During refrigeration storage condition, antioxidant activity for tea enriched yogurt is stable and it remained constant as shown in a study [2].” I think authors should mention the names of the researcher of this article (e.g. According to AUTHORS LAST NAMES [2] During refrigeration storage condition, antioxidant activity for tea enriched yogurt is stable and it remained constant).

THANK YOU FOR THE COMMENTS, AS CAN BE SEEN FROM THE TRACK CHANGES, WE HAVE SUBSTANTIALLY ALTERED THE FORMAT ACCORDING TO YOUR CONCERNS

For phase stability, authors stated in one of the answers to my comments that samples were transferred in tubes on the testing day using spatula. This should be more clearly stated in the phase stability materials and methods part, as can be confusing for the reader (in line 179 change “placing stirred ADG samples (1.0 g) in 1.5 mL tubes” for “placing (the day of measuring) stirred ADG samples (1.0 g) in 1.5 mL tubes”). Nevertheless, I find this odd, as the transferring process can affect the water holding capacity of the samples, and therefore, samples should have been stored in the 1.5ml tubes just after elaboration.

WE HAVE TRIED TO REWITE SUBSTANTIAL PARTS OF THE PAPER ON REFLECTION OF THIS COMMENT

For texture characteristics, authors do not mention when was the samples are transferred to the containers where measurements are conducted. It was the same day of measurement? or the day gels where elaborated? Please clarify  

Through the text, clarify when mentioned decrease/increase of a parameter is respect the control (e.g. in line 293 is not mentioned, among others)

Reviewer 2 Report

Thank you for the significant correction of the submitted manuscript. After corrections, the text is more readable for the reader. I suggest moving fig 1 to the supplement section.

Author Response

Thank you for your kind comments. We have revised the manuscript as you have requested as can be seen in the new paper. 

We hope that this is sufficient for publication and thank you for your help with the manuscripts.

Kind regards

Charles